# Making the Case for Standardized Outcome Measures in Exercise and Physical Activity Research in Chronic Kidney Disease

**Thomas J. Wilkinson** [1,*], **Jennifer M. MacRae** [2] , **Stephanie Thompson** [3] **and Clara Bohm** [4,5,†]
on behalf of the Global Renal Exercise Network (GREX)

1    Leicester Diabetes Centre, University of Leicester, Leicester LE5 4PW, UK
2    Cumming School of Medicine, University of Calgary, Calgary, AB T2N 1N4, Canada
3    Department of Medicine, University of Alberta, Edmonton, AB T6G 2R3, Canada
4    Department of Internal Medicine, University of Manitoba, Winnipeg, MB R3T 2N2, Canada
5    Chronic Disease Innovation Centre, Winnipeg, MB R2V 3M3, Canada
*    Correspondence: t.j.wilkinson@leicester.ac.uk
†    Membership of the Global Renal Exercise Network (GREX) is provided in the Acknowledgments Section.

**Abstract:** Physical activity and exercise are core components of lifestyle modification strategies for the management of chronic kidney disease (CKD). Yet, physical activity levels have consistently remained poor across all stages of CKD. Exercise interventions, including aerobic and resistance training, and lifestyle interventions promoting physical activity, have been shown to improve a multitude of clinical endpoints and factors important to patients; however, despite the evidence, the provision of physical activity in clinical practice is still inadequate. The usefulness of any study hinges on the adequacy and clinical relevance of the outcomes and outcome measures used. Inconsistent reporting and wide disparities in outcome use across studies limit evidence synthesis to help guide clinical practice. The kidney exercise and physical activity field has been particularly prone to inconsistent outcome reporting. To ensure research is relevant and able to influence clinical practice and future research, we need to ensure the use (and reporting) of standardized, relevant outcome measures. Core outcome sets (COS) have been widely developed across many chronic conditions, yet these COS have not been tailored to physical activity and exercise in CKD. Outcomes in clinical research need to be relevant to the intervention being employed. From this perspective, we summarize the importance that standardizing outcomes and outcome measures may have in relation to physical activity and exercise interventions for people living with kidney disease.

**Keywords:** outcomes; outcome measures; core outcome measure set; exercise; physical activity

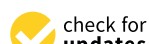



## 1. Introduction

Chronic kidney disease (CKD) is an increasingly prevalent clinical and public health problem worldwide, affecting about 8% to 16% of the general adult population, especially in people with diabetes and hypertension [1]. Kidney disease presents as a progressive disease with no cure and is characterized by high morbidity and mortality. By 2040, CKD is estimated to become the fifth leading cause of death globally, one of the largest projected increases of any major cause of death [2]. Management of CKD is complex and requires treatment of the pathophysiology of CKD itself, a multidisciplinary approach to reduce mitigating risk factors for CKD, and managing common complications and comorbidities such as cardiovascular disease, diabetes, and anemia [3]. Alongside pharmacological interventions, physical activity and exercise are considered a core component of lifestyle modification strategy in the management of CKD [4,5].

## 2. Benefits of Exercise and Physical Activity in CKD

Given the high symptom burden, reduced physical function, and multimorbidity observed in people living with CKD, it is perhaps unsurprising that physical activity levels are extremely low across all stages of CKD [6,7]. For example, in a study conducted in the UK, Wilkinson et al. [8] found physical activity levels decreased from CKD Stage 3 (17% were considered sufficiently physically active through to CKD Stages 4 and 5 (11% were active, defined using the General Practice Physical Activity Questionnaire) before reaching a nadir in people requiring dialysis (only 6% of HD and 8% of PD patients were active). Having a kidney transplant 'restored' activity levels somewhat, yet only 27% of individuals were sufficiently active. These findings are consistent with other research into physical activity levels in CKD [9–12] and are well below those of published normative values of the general population. Worryingly, many of these studies are likely subject to selection bias in that those recruited are often in better health and more favorably disposed to physical activity and health-promoting behaviors. This potentially leads to overestimates in the level of physical activity among this unrepresentative population. The association between physical inactivity and poor clinical outcomes is well established for those with CKD, those on dialysis, and in kidney transplant recipients [6]. Observational studies have consistently demonstrated an association between low levels of physical activity and higher mortality [13,14], greater incidence of disability [15], multimorbidity [8], faster decline in kidney function [16,17], higher symptom burden [18], and poorer quality of life [19].

There is now robust evidence from a wealth of recent systematic reviews and meta-analyses documenting the benefits of regular exercise and physical activity on cardiometabolic, neuromuscular, and cognitive function across all stages of CKD. Exercise interventions, including both aerobic and resistance training, and lifestyle interventions promoting physical activity, have been shown to improve a multitude of factors such as cardiorespiratory function and exercise capacity [20–24], symptoms [20,25,26], lipid profile [20,27], physical function [21,28], health-related quality of life (HRQoL) [22,24], blood pressure [24,27,29,30], and body composition variables such as muscle mass and bone health [24,27,31,32]. However, despite the benefits for disease progression risk factors, the effect of exercise on kidney function itself (e.g., eGFR, creatinine) remains less clear [33].

## 3. Diverse Heterogeneity in the Outcomes Used across Exercise and Physical Activity Research in CKD

Despite evidence from clinical trials, and recently published clinical practice guidelines [34,35] supporting physical activity and exercise in the management of CKD, implementation and recognition of exercise and physical activity as a safe and adjunct therapeutic option in nephrology remains limited across the globe [7]. Aside from the well-known threats to internal and external validity in study design, the choice of outcome (and how an outcome is measured and reported) can significantly affect the inferences derived and limit research impact [36]. Outcome measure choice is important, as the usefulness of any study as a contribution to clinical knowledge hinges on the adequacy of the outcome measure(s) used. In particular, the inconsistent reporting and wide disparity in the outcomes used across different studies limit the synthesis and pooling of data for meta-analysis. Comparing research findings to other studies increases the potential for trials to inform treatment decisions [37,38] and is an important condition for regulatory authorities and funders when evaluating novel therapies.

One only needs to look at the plethora of systematic reviews and meta-analyses of the effects of exercise and physical activity in CKD to understand the scale of the problem regarding inconsistent outcome reporting. For example, in a recent review of exercise-based interventions in kidney transplant recipients, Wilkinson et al. [21] reported a total of four broad domains such as physical fitness, quality of life, and clinical measures; within these, there existed a magnitude of specific measures. As an example, HRQoL was assessed via three distinct questionnaires, whilst measures of exercise capacity and physical function were reported from a combination of six different objective means (e.g., sit-to-stand test)

and a single subjective (e.g., Duke Activity Status Index) questionnaire. From a clinical outcome perspective, outcomes spanned ten areas, including blood pressure, inflammation, kidney function, and endothelial function.

Many clinical trials of exercise and physical activity in CKD aim to improve some element of physical fitness. Physical fitness suggests that an individual has acquired an acceptable ability needed to perform physical tasks within their current environment [39]. Whilst the reporting of physical fitness in kidney research is unclear due to the lack of definitions, particularly around the disaggregated sub-components of physical fitness (i.e., what is physical fitness), and limited recognition of the instruments and tests needed to measure outcomes, it is often seen as important by patients, carers, and health professionals [40]. Probably the most striking example of the diverse outcomes for physical fitness used in CKD trials is that by Jegatheesan et al. [41]. Here, a systematic review of 111 randomized trials (not restricted to those describing physical activity and exercise interventions) reporting physical fitness outcomes across all stages of CKD found a total of 87 tests/measurements were used to evaluate 30 outcomes measures that reported on 23 outcomes. These outcomes were categorized into five domains of physical fitness (neuromuscular fitness, exercise capacity, physiological–metabolic, body composition, and cardiorespiratory fitness). Neuromuscular fitness, the most reported domain, was examined using 37 individual tests/measurements, including the physical function component of questionnaires, one-repetition maximum tests, and hand-grip strength. The authors concluded that reducing the heterogeneity in the outcome measures used to report on physical fitness and other outcome domains in kidney disease trials, and use of consistent, appropriate, clinically relevant and patient-important outcome measures would allow for better and more consequential comparisons that may aid clinical decision making [41].

## 4. Core Outcome Sets in Nephrology

The use and reporting of standardized outcomes that directly apply to people living with CKD is fundamental in ensuring research is both applicable and able to impact clinical practice and future research [42]. Core outcome sets (COS) are defined as "agreed, standardised sets of outcomes that represent the minimum group of outcomes that should be collected and reported for all clinical trials and related evaluative research for a specific health condition" [43]. The outcomes that make up COS are chosen because they are important to all stakeholders, such as patients or policymakers, for decision making. Core outcome sets are not designed to be comprehensive or exclusive, and often include between three and five outcomes [44]. Core outcome sets have been widely developed across many chronic conditions and for distinct interventions within these conditions. They are influential in clinical research, as they aim to represent a set of outcome measures to be used in all trials. This consistency aims to reduce heterogeneity in outcomes measurement [45]. The use of COS is also increasingly being advocated by funders to ensure clinical relevance and increase the potential impact of research. Groups, such as Core Outcome Measures in Effectiveness Trials (COMET) [43], help standardize measurement across clinical research trials by providing resources to support the development of COS, including guidelines for increasing uptake and an accessible database for researchers to use.

In nephrology research, work to agree on the standardization of outcome measurement is being undertaken by the Standardized Outcomes in Nephrology (SONG) initiative [37,44]. Since its beginning in 2014, the SONG initiative has aimed to establish core outcomes across all stages of CKD through a validated and transparent process based on the Outcome Measures for Arthritis Clinical Trials (OMERACT) [46] and COMET [43] initiatives. The process used by SONG to develop COS involves systematic reviews of outcome reporting in research studies, focus groups that use structured nominal group techniques, interviews with stakeholders, international Delphi surveys, and consensus workshops [37,38]. The SONG group is not the only initiative attempting to standardize outcomes in CKD. In 2019, the International Consortium for Health Outcomes Measurement (ICHOM) also developed a standardized minimum set of patient-centered outcomes targeted for clinical use [47].

Figure 1 shows the comparison of core and essential outcomes included in SONG and ICHOM core outcome sets. At the time of writing, COS for CKD and glomerulonephritis are in development.

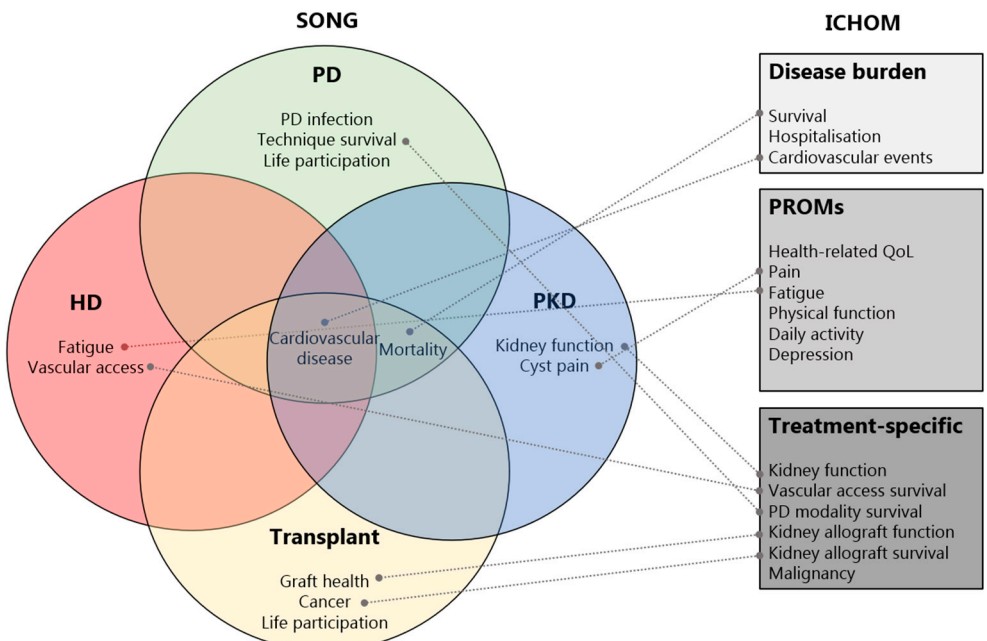

**Figure 1.** Summary of core and essential outcomes included in SONG and ICHOM adult core outcome sets. HD = Hemodialysis; PD = Peritoneal dialysis; PKD = Polycystic kidney disease; QoL = Quality of life; ICHOM = International Consortium for Health Outcomes Measurement; SONG = Standardized Outcomes in Nephrology.

Nevertheless, these COS are not focused on physical activity and exercise-based interventions for people living with kidney disease. Outcomes in clinical research need to be relevant to the intervention being employed; hence why intervention-specific COS have been developed in other fields (e.g., for hospital deprescribing trials for older people [48]). Therefore, recommended outcomes specific to exercise and physical activity research are fundamental, as these types of interventions are often performed for reasons not explicitly currently captured in SONG or ICHOM sets. For example, measurement of physical activity levels is often a key outcome in interventions designed to improve physical activity behavior; yet there remain no standardized and recommended means to assess this in CKD. In a broader sense, there is clearly some intersection between the outcomes proposed by both SONG and ICHOM, and some of these outcomes could be (and often have been) assessed in trials of physical activity and exercise (e.g., physical function, fatigue, daily activity, and kidney function). Further, some of the motives for engaging in physical activity behavior include improving QoL, life participation, control of body weight, and improvement in mood—all outcomes reported in these existing COS. Yet, there remains no consensus on how best to measure these components in the context of physical activity and exercise interventions. Overall, the lack of guidance around standardized outcomes has resulted in researchers selecting a range of diverse and discordant outcomes to measure in their studies, perpetuating the problem by further adding to the mixture of outcomes reported and reducing the impact of any trial performed.

## 5. Core Outcome Sets in Exercise and Physical Activity Research

Despite the proliferation of COS in clinical research, to date, few have been specifically developed for physical activity and exercise-based research. Table 1 outlines the only previously proposed COS for general physical activity and exercise-based research in other conditions. Interestingly, no COS work in this area was identified in any 'organ-based

diseases' (e.g., heart failure, respiratory disease). Some of these trials use a methodology that aims to identify a consensus around broad domains or areas that need to be assessed in trials, whilst others also identify specific measures that can be recommended for each of those identified domains or areas. Quality of life appears in several of the COS, including those in multiple sclerosis, musculoskeletal disorders, and general adult populations. Different means to assess quality of life were recommended, including disease-specific (e.g., Multiple Sclerosis Impact Scale) and more general instruments (e.g., EQ5D). Elements of physical function are also represented in several of the COS, along with components of self-efficacy. Some outcomes are seemingly specific to the population studied. For example, preventing falls in those with dementia. Outcomes span both physical and psychological elements, with enjoyment, purpose, and motivation featured in several COS. Compared to more clinical-based COS, hard clinical endpoints, such as mortality or hospitalization, did not feature widely, although preventing falls and utilization of health services are included in two of the studies. Despite the COS being developed for physical activity and exercise research, the measurement of physical activity and/or exercise behavior itself was only reported in two of the studies, with one study recommending the objective use of accelerometry [49].

**Table 1.** Previously proposed core outcome sets and outcome measures for physical activity and exercise-based research.

| Study | Population | Outcomes | Outcome Measures |
|---|---|---|---|
| Ramdharry et al., 2021 [50] | Rare neurological conditions | ○ Physical well-being; (ii) psychological well-being and (iii) participation in day-to-day activities <br> ○ Enjoyment, motivation, and confidence | ○ Oxford Participation and Activities Questionnaire <br> ○ Sources of Self-Efficacy for Physical Activity |
| Gonçalves et al., 2020 [51] | Dementia | ○ Preventing falls <br> ○ Doing what you can do <br> ○ Walking better, being able to stand up and climb stairs <br> ○ Staying healthy and fit <br> ○ Feeling useful and having a purpose <br> ○ Feeling brighter <br> ○ Enjoying the moment | Not identified as part of the study |
| Paul et al., 2014 [52] | Multiple sclerosis | ○ Quality of life <br> ○ Energy and drive (fatigue) <br> ○ Exercise tolerance <br> ○ Muscle function/moving around <br> ○ Body structure (body composition) | ○ Multiple Sclerosis Impact Scale (MSIS-29) or MSQoL54 <br> ○ Modified Fatigue Impact Scale (MSIF) (multi-dimensional) or Fatigue Severity Scale (FSS) (uni-dimensional) <br> ○ 6-Minute Walk Test <br> ○ Timed Up and Go <br> ○ Waist–hip ratio; body mass index (BMI) |
| Crocker et al., 2022 [49] | General adult population | ○ Device-based level of physical activity <br> ○ Health-related quality of life <br> ○ Function <br> ○ Patient satisfaction <br> ○ Physical activity | ○ Accelerometer <br> ○ EQ-5D |
| Thompson et al., 2019 [53] | Musculoskeletal disorders | ○ Quality of life <br> ○ Pain <br> ○ Cost-effectiveness <br> ○ Self-efficacy <br> ○ Knowledge to plan future exercise <br> ○ Utilization of health services | Not identified as part of the study |

The methodologies used in these studies largely followed standard guidelines, including the use of scoping literature searches, Delphi surveys, and consensus-based workshops that involved a range of different and innovative methods such as card sorting to facilitate voting in people living with dementia [51] and group discussions [52]. These studies also included a range of participants in the COS development. As well as patients, physiotherapists, occupational therapists, and nurses involved in the design or delivery of physical activity interventions were included in the study by Goncalves et al. [50]. Paul et al. [52] included experts in exercise science and physiology, along with health economics. Representatives from various neurological condition charities were involved in the study by Ramdharry et al. [50].

## 6. The Case for Standardized Outcomes in Exercise and Physical Activity Research in CKD

There has been no previous attempt to develop a standardized COS in exercise and physical activity research in CKD. Research priority exercises provide insight into questions deemed important and help enlighten us on what outcomes may be useful to measure. Using Delphi methodology, in Taryana et al. [54], patients/caregivers, researchers, clinicians, and policymakers submitted their priorities regarding research in the area of CKD and exercise. The final top research priorities included: (1) defining exercise-related outcomes that are meaningful to people living with CKD; (2) understanding the perspectives of people living with CKD on exercise; (3) identifying factors to increase motivation to participate in exercise; (4) examining the effect of exercise on the risk of institutionalization; (5) the effect of exercise on mortality and mobility for patients at all stages of CKD; (6) and understanding the effect of pre- and post-kidney transplant exercise interventions on postoperative recovery. Respondents also identified fatigue, mobility, quality of life, strength, mortality, and delay of kidney function decline as important research outcomes, but consensus on how to measure these or priority outcomes was not pursued in this study.

## 7. What Can We Do Next?

It is evident that a consensus around recommended outcomes and outcome measures will help to ensure that exercise and physical activity trials in CKD report outcomes that are directly relevant to patient care and to patient priorities. Although the broad core outcomes identified by SONG and ICHOM are important to consider when designing studies, it is likely that within the specific context of physical activity and exercise interventions, other potential outcomes may be deemed important (e.g., self-efficacy as observed in other physical activity-based COS). Physical fitness, which includes components such as exercise capacity and neuromuscular function, is necessary for participation in activities of daily living [24] and arguably a key target for many physical activity and exercise-based research—yet no consensus nor recommendations on how to measure it exist. Many of the outcomes identified in the studies in Table 1 are likely also applicable to people living with kidney disease, and there is a cross-over with the priorities identified by Taryana et al. such as quality of life, mobility, and fatigue. Any development of a COS specific to kidney disease could utilize these outcomes to form an initial longlist of outcomes to be scored and ranked for importance by key stakeholders. Further expertise and outcome recommendation could also be sought from other clinical exercise-based groups, such as the American College of Sports Medicine, or existing guidelines around physical activity in CKD, such as the 2022 'UK Kidney Association Exercise and Lifestyle in CKD Clinical Practice Guidelines' developed by Baker et al. [34].

Any development of a COS in physical activity and exercise research is not without its challenges. Any chosen outcome domain should reflect the research question, the study aims, and the target population. However, reducing the multitude of outcomes available down to a small core set may result in a COS that is 'overly blunt' [49]. As such, during the development of a COS, there must be a careful balance between measuring outcomes relevant to individual trials versus the enablement of comparisons across trials.

The previous development of COS in nephrology has navigated this balance by suggesting core or essential outcomes (i.e., those that should be measured in every trial) and outer-tier outcomes (i.e., important yet not essential). If a COS were to be developed in physical activity and exercise research, then it is likely outcomes could be separated into those that should be included in every trial and outcomes that should be reported and considered in some trials. Whether an outcome is important for all trials or just some will need to be determined by the key stakeholders involved in these types of studies (e.g., people living with kidney disease, researchers, clinicians, etc.). Following previous methodology, those deemed critically important to the majority of participants would likely be included in the core outcomes, with others classified into the outer tiers. In the development of any COS (or recommendations of outcome measures), improved COS uptake and awareness amongst researchers in the field is imperative. Numerous approaches to improve COS uptake have been identified including involving patients in COS development, adapting guidelines to support understanding of COS, and raising awareness and encouraging COS use by funders [55].

In the absence of a COS specific to physical activity and exercise interventions in CKD, we need to ensure that any outcomes we choose in physical activity and exercise research are at the very least reflective of those important to stakeholders, and as identified by SONG or ICHOM. To ensure consistency in how outcomes are being measured, it is suggested that any COS should also recommend the outcome measurement instruments needed [55]. As such, identifying standardized outcome measures of the effect of physical activity and exercise on these established core outcomes may be warranted. For example, whilst improving 'life participation' may involve a physical activity-based intervention, the identification of consistent methodology (or methodologies) to measure physical activity— either objective (e.g., accelerometry) or subjective (e.g., self-reported questionnaire)—is needed. Moreover, given that physical fitness is necessary for participation in activities of daily living [24], then the consensus behind its measurement may also be important. There are important factors that should be considered when recommending any outcome measures to be used. These include the relevant psychometric properties of the measure, the burden to the patient or person administering the measure, the appropriateness to the level of kidney disease being studies, and the specificity of the measure regarding the physical activity intervention [52]. Work by the Global Renal Exercise Network (GREX) has already begun to identify the most appropriate measures of physical functioning in this patient population, with an aim to recommend the best measures to use. However, more work is needed to identify the best measures of other important outcomes.

## 8. Conclusions

There is extensive heterogeneity in the outcomes and outcome measures used in physical activity and exercise studies in kidney disease. This has limited the ability to sufficiently synthesize evidence across research and has likely hindered the translation of research into clinical practice. Whilst COS exist in nephrology, there remains a dearth of standardization in what outcomes we should measure in physical activity and exercise interventions for people living with kidney disease, either reflective of pre-established COS or against other outcomes important to patients that might not be already considered. Outcomes in clinical research need to be relevant to the intervention being employed. We argue that a minimum set of measures would help bring consistency in reporting and permit greater comparisons across studies that can help inform better clinical decisions.

**Author Contributions:** Conceptualization, T.J.W., J.M.M., S.T. and C.B.; writing—original draft preparation, T.J.W.; writing—review and editing, J.M.M., S.T. and C.B. All authors have read and agreed to the published version of the manuscript.

**Funding:** This research received no external funding.

**Institutional Review Board Statement:** Not applicable.

**Informed Consent Statement:** Not applicable.

**Data Availability Statement:** Not applicable.

**Acknowledgments:** This study is endorsed by the Global Renal Exercise Network (GREX). The interpretation and conclusions contained herein are those of the researchers and do not represent the views of GREX. All authors (T.J.W., J.M.M., S.T., C.B.) are members of GREX, which aims to foster research and innovation across multiple disciplines to develop strategies to increase physical activity and improve health outcomes for people with CKD (https://grexercise.kch.illinois.edu/, accessed on 9 May 2023).

**Conflicts of Interest:** The authors declare no conflict of interest.

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
