# Peer review of "Making the Case for Standardized Outcome Measures in Exercise and Physical Activity Research in Chronic Kidney Disease"

_kidneydial, doi:10.3390/kidneydial3020020_

Round 1

Reviewer 1 Report

I appreciate the opportunity to review the perspective manuscript: Making the case for standardized outcome measures in exercise and physical activity research in chronic kidney disease. Exercise effects on chronic disease do not get enough attention in the prevention and progression slowing effects it enacts in those with chronic kidney disease. The choosing of effective measures and tailoring those measures to a patient can be a delicate process and the authors bring the current issues of hetereogenetiy of these measure to the limelight.

I feel the article is well written and describes the current issues well as well as the difficulty of the current research to synthesize clinical practice guidelines using various standardization measures. The authors also do a good job in providing frameworks that may support the formation of such guidelines and do not provide possible solutions thereby keeping within the scope of the article format and does not over-reach which I applaud. 

Major comments

I have no major concerns with this article. 

Minor comments

I feel the authors could have recommended not only the frameworks, which they did quite well with, but also provided an idea on recommending other working groups, professional expertise when starting to create these COS. For example, the American College of Sports Medicine professional come to mind, especially from their clinical section. I do recognize the idea of stakeholders, patients, physicians, so using a community-base participatory research approach could be acknowledged as well. 

line 219 "are other potentially other outcomes.." please reword sentence, ..other potential outcomes..

Reviewer 2 Report

  1. Line 33

””Kidney disease presents as a progressive disease with no cure and characterized by high morbidity and mortality.”

That may be unconventional.

I think the current nephrological research progress is ongoing.   Until now, therapeutic interventions have been limited, but new treatments are being actively developed nowadays, and there are many discussions about setting outcomes in clinical trials. In addition, dialysis technology has made remarkable progress, and there is a high possibility that dialysis technology will spread rapidly in developing countries in the future.

  1. Line 58

Management of CKD is complex and requires treatment of CKD itself, a multidisciplinary approach to reduce risk factors for CKD and managing common complications such as cardiovascular disease, diabetes and anemia. [

Additional description is necessary that the cause or pathophysiology of CKD itself also becomes the therapeutic target.

3.

 It has little difficulty to understand the importance of  selection bias.

Does it mean that participating patients have better basic physical and cognitive functions so that exercise can be continued in study periods, or are there other selection biases?

4.

Line85 Does inconsistent outcomes mean looking across different research papers? .

5.

Line 182

 If  there are any references for other organ diseases other than musculoskeletal diseases such as respiratory diseases and heart diseases?

The concept of  outcome should be considered as problems of the common  chronic organ damage and  target of intervention. It will further clarify the authors’ suggestion.

I have annotated the manuscript with several minor recommendation above, which I believe will improve the readability and help understanding of the paper.

Reviewer 3 Report

Overall, the abstract should be revised to clarify the research question and highlight the specific contributions of the paper. The abstract could benefit from an introductory sentence that succinctly summarizes the main point of the paper, followed by a brief explanation of the context and the importance of the topic. The abstract should also provide more details on the specific outcomes and outcome measures that the authors recommend using in physical activity and exercise interventions for CKD.

"multitude of clinical endpoints and factors important to patients; however, despite the evidence," instead of "contrary to the evidence."

Line 15 should read "multitude of clinical endpoints and factors important to patients; however, despite the evidence," instead of "contrary to the evidence."

Line 18 should read "Inconsistent reporting and wide disparities in outcome use across studies limit evidence synthesis to help guide clinical practice."

Line 20 should read "The kidney exercise and physical activity field has been particularly prone to inconsistent outcome reporting."

Line 23 should read "yet these COS have not been tailored to physical activity and exercise in CKD."

Line 25 should read "we summarize the importance that standardizing outcomes and outcome measures may have on physical activity and exercise interventions for people living with CKD."

 In line 39, it might be better to use the phrase "lifestyle modification strategy" instead of "any lifestyle modification strategy."

In lines 45-48, the phrase "defined as 'active' on the General Practice Physical Activity Questionnaire" could be moved to line 46 to make the sentence clearer.

In line 52, it would be useful to explain what is meant by "selection bias" and how it might affect physical activity estimates.

In line 69, it would be helpful to clarify what is meant by "kidney function itself" and how it differs from disease progression risk factors.

Line 141: "all stages with CKD" should be "all stages of CKD."

For suggestions, the statement in line 180 could be revised to provide more context on how the diverse and discordant range of outcomes perpetuates the problem. Additionally, the article could benefit from more in-depth analysis and comparison of the previously proposed COS in Table 1 and how they could be adapted for exercise and physical activity research in people living with kidney disease.

In line 186, "agreement around broad domains" should be revised to "consensus around broad domains".

In line 233, "should" should be corrected to "could" to reflect the possibility of outcomes being separated into essential and important tiers.

In line 236, "improving" should be corrected to "improved" to reflect the past tense of the sentence.

In line 243, "thr" should be corrected to "the."

In line 245, "measurement" should be pluralized to "measurements."

Suggestion: In line 203, "exercise-related outcomes" could be more clearly defined, as it may not be immediately clear what is meant by this term. Additionally, in line 234, it could be helpful to provide guidance on how to determine which outcomes should be included in every trial and which should be considered in some trials.

Round 2

Reviewer 3 Report

I reviewed the revised manuscript and the response to minor reviewers' comments. Revised Manuscript is well written. All comments have been addressed and thus accepted for publication.